DOI: 10.1038/ncomms11617　　OPEN

# Microscopic origins of the terahertz carrier relaxation and cooling dynamics in graphene

Momchil T. Mihnev[1,2,*], Faris Kadi[3,*], Charles J. Divin[1,2], Torben Winzer[3], Seunghyun Lee[1,4], Che-Hung Liu[1], Zhaohui Zhong[1], Claire Berger[5,6], Walt A. de Heer[5,7], Ermin Malic[3,8], Andreas Knorr[3] & Theodore B. Norris[1,2]

The ultrafast dynamics of hot carriers in graphene are key to both understanding of fundamental carrier–carrier interactions and carrier–phonon relaxation processes in two-dimensional materials, and understanding of the physics underlying novel high-speed electronic and optoelectronic devices. Many recent experiments on hot carriers using terahertz spectroscopy and related techniques have interpreted the variety of observed signals within phenomenological frameworks, and sometimes invoke extrinsic effects such as disorder. Here, we present an integrated experimental and theoretical programme, using ultrafast time-resolved terahertz spectroscopy combined with microscopic modelling, to systematically investigate the hot-carrier dynamics in a wide array of graphene samples having varying amounts of disorder and with either high or low doping levels. The theory reproduces the observed dynamics quantitatively without the need to invoke any fitting parameters, phenomenological models or extrinsic effects such as disorder. We demonstrate that the dynamics are dominated by the combined effect of efficient carrier–carrier scattering, which maintains a thermalized carrier distribution, and carrier–optical–phonon scattering, which removes energy from the carrier liquid.

[1] Department of Electrical Engineering and Computer Science, University of Michigan, Ann Arbor, Michigan 48109, USA. [2] Center for Ultrafast Optical Science, University of Michigan, Ann Arbor, Michigan 48109, USA. [3] Institut für Theoretische Physik, Nichtlineare Optik und Quantenelektronik, Technische Universität Berlin, D-10623 Berlin, Germany. [4] Department of Electronics and Radio Engineering, Kyung Hee University, Gyeonggi 446-701, South Korea. [5] School of Physics, Georgia Institute of Technology, Atlanta, Georgia 30332, USA. [6] Institut Neel, CNRS UJF-INP, Grenoble 38042, Cedex 6, France. [7] King Abdulaziz University, Jeddah 22254, Saudi Arabia. [8] Department of Applied Physics, Chalmers University of Technology, SE-412 96 Gothenburg, Sweden. * These authors contributed equally to this work. Correspondence and requests for materials should be addressed to T.B.N. (email: tnorris@umich.edu).

In graphene, a linearly polarized ultrafast optical pulse excitation gives rise to an initially anisotropic distribution of carriers at high energies[1,2]. Efficient carrier–carrier and carrier–phonon interactions quickly relax the hot carriers to an isotropic thermal distribution, which is then followed by carrier cooling as energy is transferred from the carrier population to the lattice[3–7]. A schematic illustration of these processes is given in Fig. 1 (see Supplementary Note 1). A variety of dynamical phenomena, such as the appearance of significant carrier multiplication and transient optical gain, have been theoretically predicted[6,8–10] and experimentally demonstrated[6,11,12]. The dynamics observed by a time-domain terahertz (THz) probe pulse, however, exhibit a number of features that have been interpreted in the framework of phenomenological models, but as yet have not been understood quantitatively or qualitatively in terms of fundamental microscopic many-particle processes. For example, the observed photoinduced THz conductivity may be either positive or negative. The positive photoinduced THz conductivity has been viewed in the context of simple Drude models as stemming from enhanced free-carrier intraband absorption upon photoexcitation[13,14], while the negative photoinduced THz conductivity has been attributed variously to stimulated THz emission[15], enhanced carrier scattering with optical phonons, surface optical phonons or charge impurities[16–18] and carrier heating[19–21].

It is generally understood that the initial cooling of hot thermalized carriers proceeds via the emission of high-energy optical phonons ($\hbar\omega_{op} \approx 200$ meV). Once the carrier temperature is sufficiently below the optical phonon energy, the cooling should proceed via the emission of low-energy acoustic phonons ($\hbar\omega_{ac} \approx 4$ meV). For graphene samples with low doping density, the cooling of hot carriers near the Dirac point is expected to be very slow due to the vanishing density of states, the energetic mismatch with the energy of optical phonons, and the weak scattering with acoustic phonons[4,22,23]. The slowest observed cooling times, however, have been on the order of hundreds of picoseconds[14], much shorter than the few nanoseconds expected from carrier–acoustic–phonon scattering in ideal graphene[22,23], leading to proposals that the cooling is dominated by disorder-assisted electron–phonon (supercollision) scattering[24]. The central role of disorder in these models implies that the underlying enhancement of carrier–acoustic–phonon scattering should strongly depend on the quality and the degree of disorder of the particular graphene sample. Hence, both a methodical experimental investigation and a microscopic theoretical treatment are markedly needed to provide a rigorous foundation for understanding the THz dynamics of hot carriers in graphene.

In the following, we present the results of a systematic experimental study of the THz carrier dynamics in a wide variety of graphene samples, using ultrafast time-resolved THz spectroscopy[25], in which we vary the graphene fabrication method, the number of graphene layers and their stacking orientation, the degree of disorder, the Fermi level, the carrier temperature, the substrate temperature and the type of underlying substrate to determine the dominant mechanisms responsible for the hot-carrier relaxation and cooling dynamics for different graphene material parameters and under different experimental conditions. The experimental programme is complemented by the development of a theoretical model[26,27] based on the density-matrix formalism that provides microscopic access to the time- and momentum-resolved dynamics of the carrier occupation, the phonon population for different optical and acoustic phonon modes, and the microscopic polarization determining the optical excitation of a disorder-free graphene system. An essential component of the theory is that it incorporates explicitly the time-dependent response of the system to the THz probe pulse; the macroscopic intraband current density induced by the THz

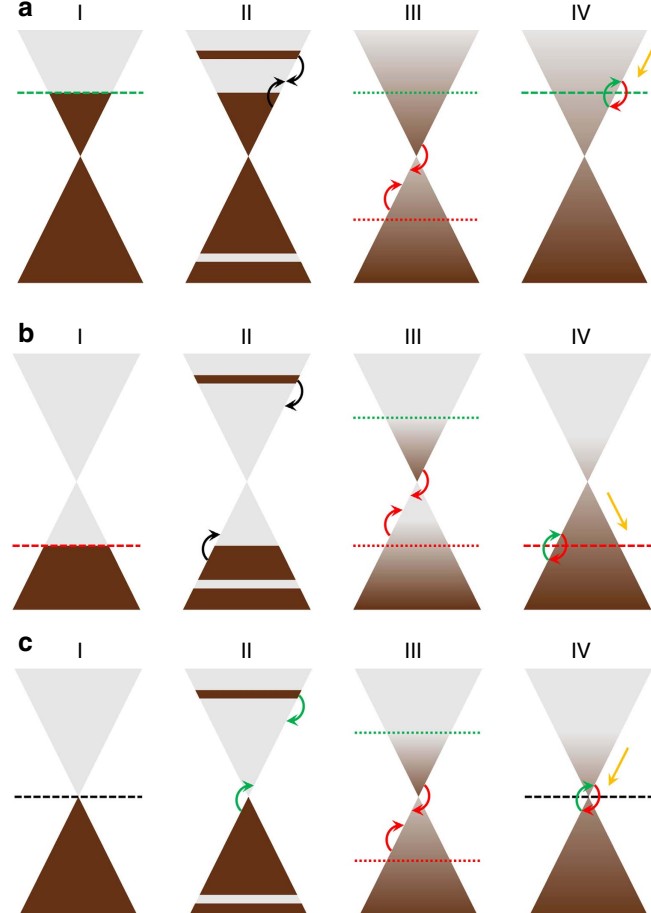

**Figure 1 | Hot-carrier relaxation and cooling dynamics in graphene with various doping densities.** (**a,b**) Hot-carrier relaxation and cooling dynamics in n-type (**a**) and p-type (**b**) highly doped graphene (step I). Initially, the optical pump pulse injects hot non-equilibrium carriers at high energies (step II). The hot electrons and holes thermalize within the conduction and valence bands, respectively, due to very efficient intraband carrier–carrier scattering processes (step III). As the hot carriers relax to lower energies, interband Auger recombination processes become allowed that quickly merge the separate electron and hole quasi-Fermi levels and lead to a single uniform hot-carrier Fermi-Dirac distribution within ~100–200 fs after photoexcitation[3,6,7] (step IV). The hot carriers cool further via optical phonon emission facilitated by very efficient carrier–carrier rethermalization in highly doped graphene. (**c**) Hot-carrier relaxation and cooling dynamics in undoped (very lightly doped) graphene (step I). Initially, the optical pump pulse injects hot non-equilibrium carriers at high energies (step II). In contrast to **a** and **b** interband impact ionization processes are possible and lead for a moderate excitation regime to significant carrier multiplication in the conduction band. Similar to **a** and **b** as the hot carriers relax to lower energies, interband Auger recombination processes become allowed (step III) that lead to a single uniform hot-carrier Fermi-Dirac distribution within ~100–200 fs after photoexcitation[3,6,7] (step IV). The hot carriers cool further via optical phonon emission, which becomes increasingly inefficient at low carrier temperatures due to the small phase space near the Dirac point as the high-energy tail of the hot-carrier distribution diminishes asymptotically. At low carrier temperatures, acoustic phonon emission and/or other slow cooling processes can also have a contribution to hot-carrier cooling in some graphene samples[22–25].

probe, and hence the resulting dynamic THz response, is calculated by microscopically accounting for the full time- and momentum-dependent carrier–carrier and carrier–phonon interactions. We find that this first-principles microscopic

approach explains completely all results without the need for any fitting parameters, phenomenological models or extrinsic effects such as disorder, which strongly suggests that the role of supercollisions in the hot-carrier dynamics has been largely overstated in the literature[24,28,29]. Moreover, the theory allows us to go beyond idealized Drude models for the dynamic THz response, and determine the limitations of such simplified models[13–21]. This work establishes that the hot-carrier dynamics are governed by the coupling of extraordinarily efficient carrier–carrier and carrier–optical–phonon interactions. This is in sharp contrast to electrical transport, which in some graphene samples is dominated by interactions with defects, charge impurities, breaks and ripples (that is, extrinsic effects) since carriers move at the Fermi energy[30–33].

## Results

**Graphene samples**. The specific graphene samples in this study include multilayer epitaxial graphene (MEG), which is grown on the C-face of single-crystal 4H-SiC(000$\bar{1}$) substrates by thermal decomposition of Si atoms[34,35], single-crystal chemical-vapor-deposited (CVD) graphene (sCVDG), which is grown on oxygen-rich copper foil into large individual single crystals exceeding hundreds of micrometres in size[36], and polycrystalline CVD graphene (pCVDG), which is grown on regular copper foil into large continuous layers with domain sizes on the order of hundreds of nanometres[37]. The as-grown CVD graphene samples are transferred to various substrates for the THz measurements. Because the different graphene samples are synthesized using completely different techniques, their doping density and degree of disorder are also very different (see Methods section), which allows us to observe the possible impact of the Fermi level and the degree of disorder on the dynamic THz response.

**Ultrafast time-resolved THz spectroscopy experiment**. To study the dynamic THz response of graphene, we utilize ultrafast time-resolved THz spectroscopy[25,38,39], which has established itself as a powerful all-optical experimental technique for directly probing the relaxation and cooling dynamics of photoexcited carriers (see Methods section). A 60-fs ultrafast optical pump pulse at 800 nm wavelength injects hot carriers into the graphene sample. The dynamic THz response is monitored by a single-cycle THz probe pulse at a variable time delay after the optical pump. The transmitted THz probe is detected using time-domain electro-optic sampling and frequency-domain THz spectra are obtained via Fourier transformation of the time-domain THz electric field. The THz carrier dynamics are acquired by monitoring the THz transmission only at the peak of the THz probe pulse, while varying the pump–probe delay. The differential THz transmission signal, $\Delta t/t$, is the change in the THz probe transmission through the graphene sample due to photoexcitation by the optical pump, normalized to the THz transmission without photoexcitation.

We begin by summarizing the main features of the data. First, we consider MEG, sCVDG and pCVDG samples having high doping density ($|\varepsilon_F| \sim 100$–$400$ meV). Figure 2a,b shows representative differential THz transmission signals as a function of pump–probe delay, for variable pump fluence and for variable substrate temperature, respectively, for a MEG sample with three layers. The secondary peak in the $\Delta t/t$ signal at $\sim 7$ ps is due to a round-trip reflection of the optical pump inside the substrate that photoexcites additional carriers. Similar results are obtained for the other two types of graphene samples (see Supplementary Figs 1–5, Supplementary Table 1 and Supplementary Notes 2–4). Based on extensive measurements on many graphene samples under various experimental conditions, we find that the THz carrier dynamics in all highly doped graphene samples are

strikingly similar. The $\Delta t/t$ signal is positive under all experimental conditions, which corresponds to a pump-induced increase of the THz transmission or a decrease of the THz absorption.

Phenomenological fits to the data in Fig. 2a,b (dashed black lines) reveal that the differential THz transmission follows closely a mono-exponential relaxation under all experimental conditions. A summary of the extracted carrier relaxation times as a function of pump fluence and substrate temperature for the highly doped graphene samples is presented in Fig. 3a,b. We note that the relaxation times of all highly doped graphene samples are weakly dependent on the pump fluence and completely independent of the substrate temperature. In addition, they are very similar in value and in the range of $\sim 1$–$3$ ps with sample-to-sample variation within $\sim 20$–$30$% despite the wide range of disorder present in the array of samples studied. On average, MEG samples exhibit slightly longer relaxation times than sCVDG and pCVDG samples, which can be attributed to a degree of disorder arising from charge impurities, substrate roughness, wrinkling and breaking of the transferred CVDG samples that can provide additional parallel channels for carrier cooling. We note, however, that the relaxation times of some CVDG samples can approach or exceed these of MEG samples, indicating that disorder-assisted electron–phonon (supercollision) cooling[24,28,29] is not the dominant cooling mechanism in our high-quality graphene samples, but generally provides at most only a modest correction.

Figure 2c,d shows the THz carrier dynamics calculated within the microscopic theory for disorder-free highly doped graphene ($|\varepsilon_F| = 300$ meV) under similar experimental conditions (see Methods section) and Fig. 3c,d shows the calculated carrier relaxation times to directly compare with the experiments. We observe that the theory is in excellent agreement with experiment, and reproduces the weak pump fluence dependence and the complete substrate temperature independence. In sharp contrast, the supercollision model[24,28,29] totally fails to capture the substrate temperature independence. The measured THz carrier dynamics can be completely reproduced neglecting carrier–acoustic–phonon scattering in the microscopic theory. Hence, the hot-carrier dynamics are directly the result of an interplay between efficient carrier–carrier and carrier–optical–phonon scattering; optical phonon emission removes energy from the high-energy tail of the hot-carrier distribution, which is maintained by efficient carrier–carrier rethermalization. The physical reason the differential THz transmission of highly doped graphene is independent of the substrate temperature is that the equilibrium THz transmission itself is insensitive to it, when the substrate temperature is far below the Fermi temperature. The slight increase of the relaxation times with increasing pump fluence is due to the re-absorption of hot optical phonons generated during the initial carrier thermalization.

The THz carrier dynamics in graphene samples having very low doping density are very different. We consider specifically MEG samples in which only the first few layers closest to the underlying SiC substrate have high doping density ($|\varepsilon_F| \sim 100$–$400$ meV) and the large number of top layers have very low doping density ($|\varepsilon_F| \sim 10$ meV). In these MEG samples, the many top lightly doped layers completely dominate the measured THz carrier dynamics. Figure 4a,b shows representative differential THz transmission signals for variable pump fluence and for variable substrate temperature, respectively, for a MEG sample with 63 layers. The secondary dip in the $\Delta t/t$ signal at $\sim 7$ ps is again due to a round-trip substrate reflection of the optical pump. In sharp contrast to the high doping case, the $\Delta t/t$ signal is negative under all experimental conditions, which corresponds to a pump-induced decrease of the THz transmission or an increase of the THz absorption.

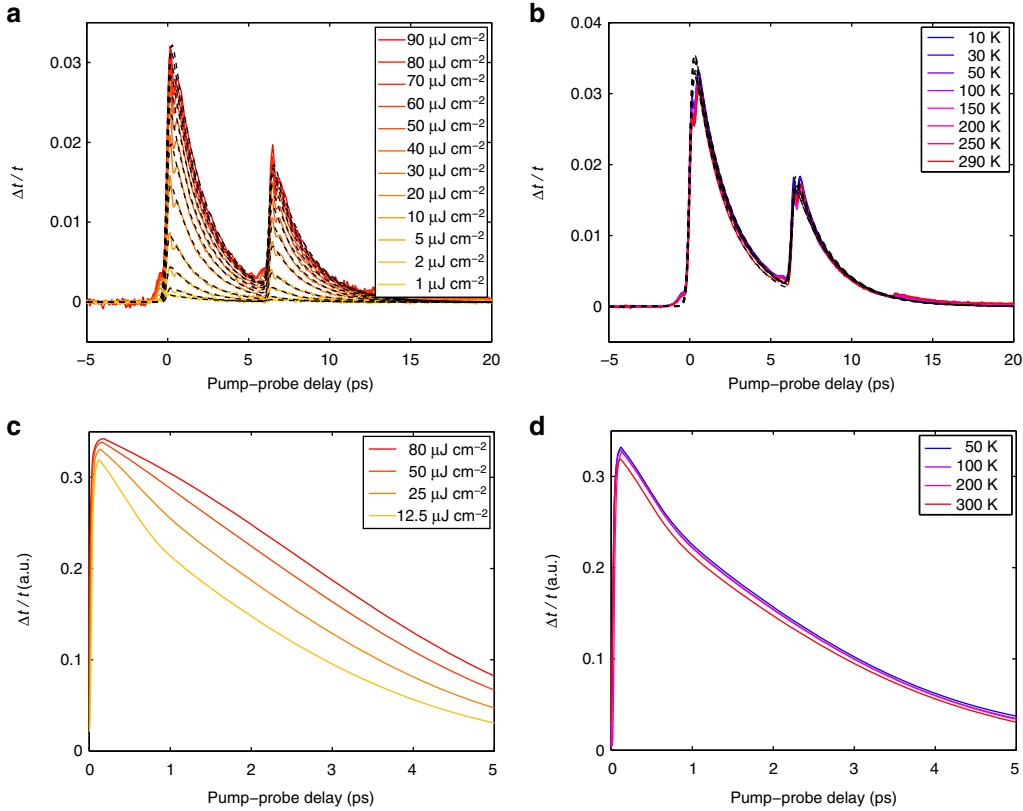

**Figure 2 | THz carrier dynamics in graphene with high doping density.** (**a,b**) Experimental differential THz transmission, $\Delta t/t$, as a function of pump–probe delay recorded at a substrate temperature of 300 K for a few different pump fluences (**a**) and at a pump fluence of 60.0 µJ cm$^{-2}$ for a few different substrate temperatures (**b**) for a highly doped MEG sample with three layers. The THz carrier dynamics follow a fast mono-exponential relaxation at all substrate temperatures and all pump fluences (dashed black lines). (**c,d**) Theoretical differential THz transmission, $\Delta t/t$, as a function of pump–probe delay calculated within the microscopic theory at a substrate temperature of 300 K for a few different pump fluences (**c**) and at a pump fluence of 12.5 µJ cm$^{-2}$ for a few different substrate temperatures (**d**) for disorder-free highly doped graphene ($|\varepsilon_F| = 300$ meV). Experiment and theory are in excellent agreement under all conditions.

Phenomenological fits to the data in Fig. 4a,b (dashed black lines) reveal that the differential THz transmission evolves from a faster mono-exponential relaxation at room temperature to a slower bi-exponential relaxation at cryogenic temperatures. A summary of the extracted carrier relaxation times as a function of pump fluence at room temperature for two MEG samples with 35 and 63 layers is presented in Fig. 5a. We note that the relaxation times at room temperature do not depend on the number of layers, which supports the interpretation that the measured THz carrier dynamics are dominated by the many top lightly doped layers. This conclusion is further supported by the fact that the maximum $\Delta t/t$ signal scales roughly linearly with the number of layers (see Supplementary Figs 3–5 and Supplementary Notes 3–4). In addition, the relaxation times of lightly doped graphene samples are in the range of ~4–7 ps, which is longer than the relaxation times of highly doped graphene samples (see Fig. 3a); this is due to a reduced efficiency of carrier–carrier scattering, which scales with carrier density. For cryogenic temperatures, we extract both short and long carrier relaxation times from the bi-exponential fits, which we associate with two distinct cooling mechanisms. As explained below, the fast carrier relaxation component on a timescale of tens of picoseconds is fully accounted for by the combined effect of carrier–carrier and carrier–optical–phonon scattering in the absence of disorder. Figure 5b shows the short relaxation times as a function of substrate temperature for variable pump fluence for the MEG sample with 63 layers. We note that the short relaxation times of

all lightly doped graphene samples are weakly dependent on the pump fluence, but strongly dependent on the substrate temperature.

Figure 4c,d shows the THz carrier dynamics calculated within the microscopic theory for disorder-free undoped graphene ($|\varepsilon_F| = 0$ meV) under similar experimental conditions (see Methods section) and Fig. 5c,d shows the calculated carrier relaxation times. The theory captures again all trends observed in the experiments. As in the high-doping case, the hot-carrier relaxation occurs via the interplay between efficient carrier–carrier and carrier–optical–phonon scattering. In contrast to the high-doping case, however, the relaxation times increase significantly with decreasing substrate temperature. This behaviour for undoped graphene is due to the substrate temperature dependence of the equilibrium THz transmission and not the dynamic non-equilibrium part of the differential THz transmission; that is, the initial Fermi surface, where the THz probe pulse acts on the carriers, strongly depends on the substrate temperature, when it is comparable to the Fermi temperature. Similar to the high-doping case, the theory shows a slight increase of the relaxation times with increasing pump fluence that can be traced back to hot phonon effects.

At low substrate temperatures, the THz carrier dynamics in the MEG samples with many lightly doped layers relax on a timescale exceeding hundreds of picoseconds, corresponding to the slow carrier relaxation component in the bi-exponential decay. As the high-energy tail of the hot-carrier distribution diminishes

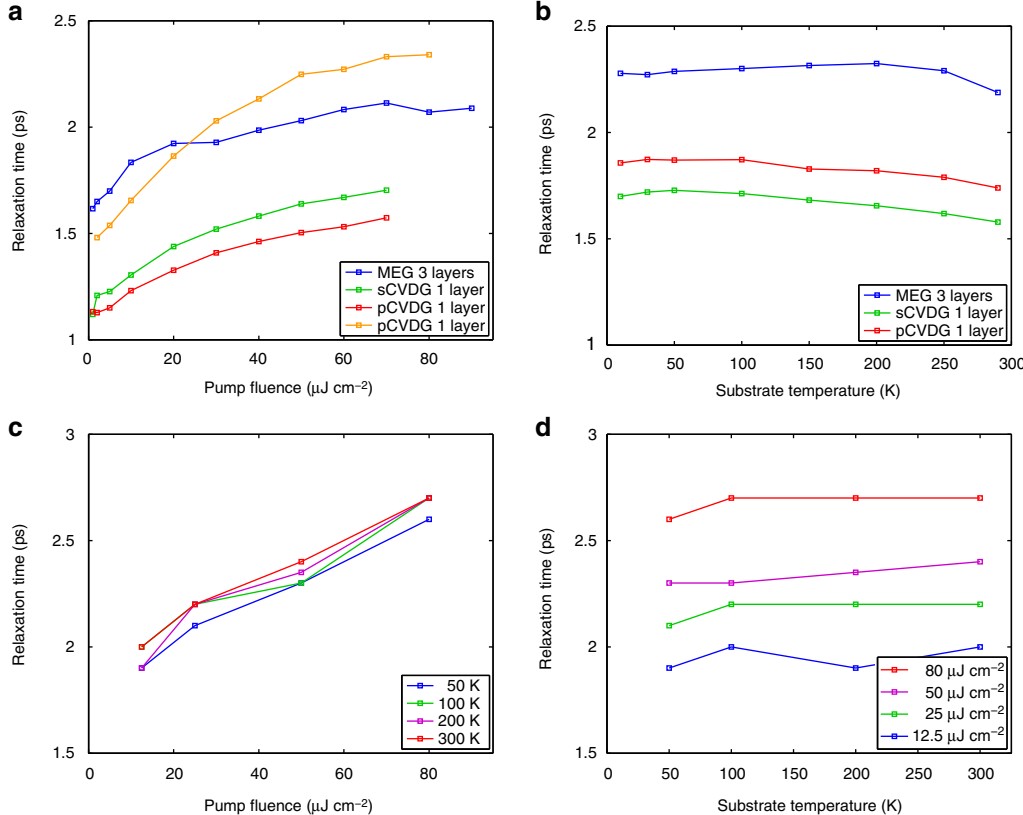

**Figure 3 | THz carrier relaxation times in graphene with high doping density.** (**a,b**) Carrier relaxation times extracted from fits to experimental differential THz transmission, $\Delta t/t$, as a function of pump fluence at a substrate temperature of 300 K (**a**) and as a function of substrate temperature at a pump fluence of 60.0 μJ cm$^{-2}$ (**b**) for highly doped MEG, sCVDG and pCVDG samples. (**c,d**) Carrier relaxation times extracted from fits to theoretical differential THz transmission, $\Delta t/t$, as a function of pump fluence for a few different substrate temperatures (**c**) and as a function of substrate temperature for a few different pump fluences (**d**) for disorder-free highly doped graphene ($|\varepsilon_F| = 300$ meV). The theory accurately reproduces the magnitude of the relaxation times and the trends with pump fluence and substrate temperature observed in the experiments.

asymptotically, a different cooling mechanism becomes dominant below ∼200 K. We have analysed the physical origin of this mechanism in a separate publication[25]. In particular, the THz carrier dynamics become dependent on the number of layers in the MEG sample, indicating that interlayer thermal coupling effects are important.

**Microscopic theory**. We now turn to a discussion of the theoretical approach used to calculate the dynamic THz response described above (see Methods section, Supplementary Fig. 6 and Supplementary Note 5). The theory is an extension of the methods developed previously by some of the authors[1,2,5,9,11,12,26,27], and is extended in this work to rigorously include the effect of the time-dependent THz probe electric field. By selectively switching on and off the different scattering processes in the model, we find that the acoustic phonon modes have negligible contribution to the observed THz carrier dynamics on the timescale of tens of picoseconds; the observed dynamics are fully accounted for by the combined effect of carrier–optical–phonon scattering (which transfers energy from the carrier liquid to the lattice) and carrier–carrier scattering (which continuously rethermalizes the carrier population as high-energy carriers lose their energy to the lattice). As seen in the comparisons between experiment and theory above, this first-principles microscopic approach explains completely all experimental results without the need for any fitting

parameters, phenomenological models or extrinsic effects such as disorder.

The standard Drude model, which is often employed as a phenomenological basis for the interpretation of graphene transport and optical data, can be obtained by assuming a constant time- and momentum-independent scattering rate in the microscopic theory (see Methods section and Supplementary Note 6). It is thus interesting to consider to what extent the Drude model can account for the observed dynamics. Figure 6a shows the pump-induced temporal evolution of the carrier temperature $T(t)$ and the Fermi level $\varepsilon_F(t)$ obtained by solving the full graphene Bloch equations within this approximation for low (12.5 μJ cm$^{-2}$ (red line)) and high (80 μJ cm$^{-2}$ (blue line)) photoexcitation, for disorder-free highly doped graphene ($|\varepsilon_F| = 300$ meV). For very high carrier temperature, as one has at early time delay and high fluence, the calculated $\Delta t/t$ signal is negative, in contradiction to the experimental observation that the $\Delta t/t$ signal is positive at all time delays for highly doped graphene samples. Only after the carrier temperature drops below ∼2,200 K does the $\Delta t/t$ signal become positive. If one makes the further approximation (as is often done in Drude models) that the only effect of the optical pump is to heat the carriers, that is, the Fermi level is assumed to remain constant, than the calculated $\Delta t/t$ signal is negative at all time delays (dashed black line in Fig. 6a). The full microscopic theory including the transient carrier temperature $T(t)$, the transient Fermi level $\varepsilon_F(t)$ and the explicitly time- and momentum-dependent scattering rates

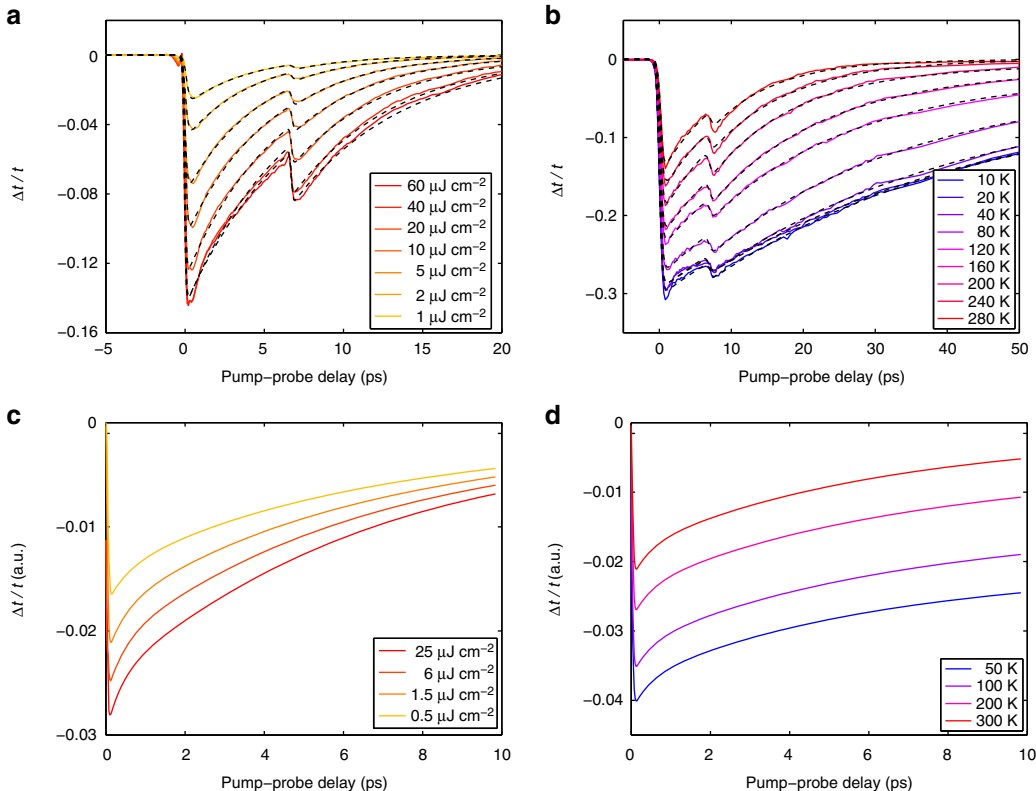

**Figure 4 | THz carrier dynamics in graphene with low doping density.** (**a**,**b**) Experimental differential THz transmission, $\Delta t/t$, as a function of pump–probe delay recorded at a substrate temperature of 300 K for a few different pump fluences (**a**) and at a pump fluence of 23.4 μJ cm$^{-2}$ for a few different substrate temperatures (**b**) for a lightly doped MEG sample with 63 layers. The THz carrier dynamics evolve from a faster mono-exponential relaxation at room temperature to a slower bi-exponential relaxation at cryogenic temperatures (dashed black lines). (**c**,**d**) Theoretical differential THz transmission, $\Delta t/t$, as a function of pump–probe delay calculated within the microscopic theory at a substrate temperature of 300 K for a few different pump fluences (**c**) and at a pump fluence of 1.5 μJ cm$^{-2}$ for a few different substrate temperatures (**d**) for disorder-free undoped graphene ($|\varepsilon_F| = 0$ meV). Experiment and theory are in excellent agreement under all conditions.

$\Gamma_{\lambda k}^{in/out}(t)$, on the other hand, completely reproduces all the features of the experimental data under all conditions. Drude models are not sufficient to consistently explain all the data, particularly the behaviour at higher pump fluence, for which the short time dynamics would exhibit a negative $\Delta t/t$ signal, unless ad hoc phenomenological parameters such as a carrier heating efficiency[19,20] or a non-monotonic carrier-temperature-dependent Drude weight[21] are added to the model. We conclude that a Drude model approximation to the full microscopic theory can provide only a semi-qualitative framework for interpreting experimental data in highly doped and undoped graphene samples at low photoexcitation; however, the full microscopic theory is required to explain the data consistently for all photoexcitation levels.

The microscopic theory also allows us to address an important question in hot-carrier physics, namely, the energy dependence of the carrier scattering rate (inverse time), defined here as the exponential decay of the carrier occupation (see Methods section). For weak THz probe excitations (as is the case here), the Boltzmann equation reveals a direct relation between the microscopic scattering rates and the exponential decay of the carrier occupation. By fitting the numerically calculated hot-carrier dynamics, we obtain the energy relaxation time of the photoexcited carriers, named the carrier scattering time. Figure 6b shows the calculated carrier scattering time $\tau(\varepsilon)$ as a function of the excess carrier energy $\varepsilon$ for disorder-free undoped graphene ($|\varepsilon_F| = 0$ meV) at a substrate temperature of 300 K. We observe that the carrier scattering time is precisely

inverse to the carrier energy, $\tau(\varepsilon) = \beta/|\varepsilon|$ (with $\beta \approx 0.9$ eV ps), over a very broad energy range ($|\varepsilon| \sim 0.2$–1.5 eV) in agreement with previous experimental studies on MEG[40] and graphite[41,42]. This behaviour is a direct consequence of the linear density of states and is therefore independent of the substrate temperature. For highly doped graphene, a deviation from the strictly inverse relation can be expected for $\varepsilon \approx \varepsilon_F$. However, our microscopically determined energy dependence of the carrier scattering time is in sharp contrast with the linear relation on the carrier energy, $\tau(\varepsilon) = \alpha|\varepsilon|$ (refs 30,31), that has been inferred from electrical transport measurements in some graphene samples. This is attributed to the fact that in electrical transport measurements carriers have energies close to the Fermi energy, and carrier scattering in these graphene samples is dominated by extrinsic mechanisms such as defects, charge impurities, breaks and ripples[32,33] likely introduced during the multistep synthesis, transfer and fabrication processes. Transferred graphene (for example, on SiO$_2$) has local spatial charge inhomogeneities under overall charge-neutral conditions due to disorder, charge impurities or surface corrugation[43–45], which obscure the low-energy graphene band structure and prevent one from studying the true graphene physics near the Dirac point. Such effects can be minimized by placing the graphene on ultra-smooth substrates such as hexagonal BN (h-BN)[46] or by suspending it[47]. The lightly doped layers in MEG are naturally protected, which makes them an ideal graphene system for studying the carrier dynamics within a few meV of the Dirac point.

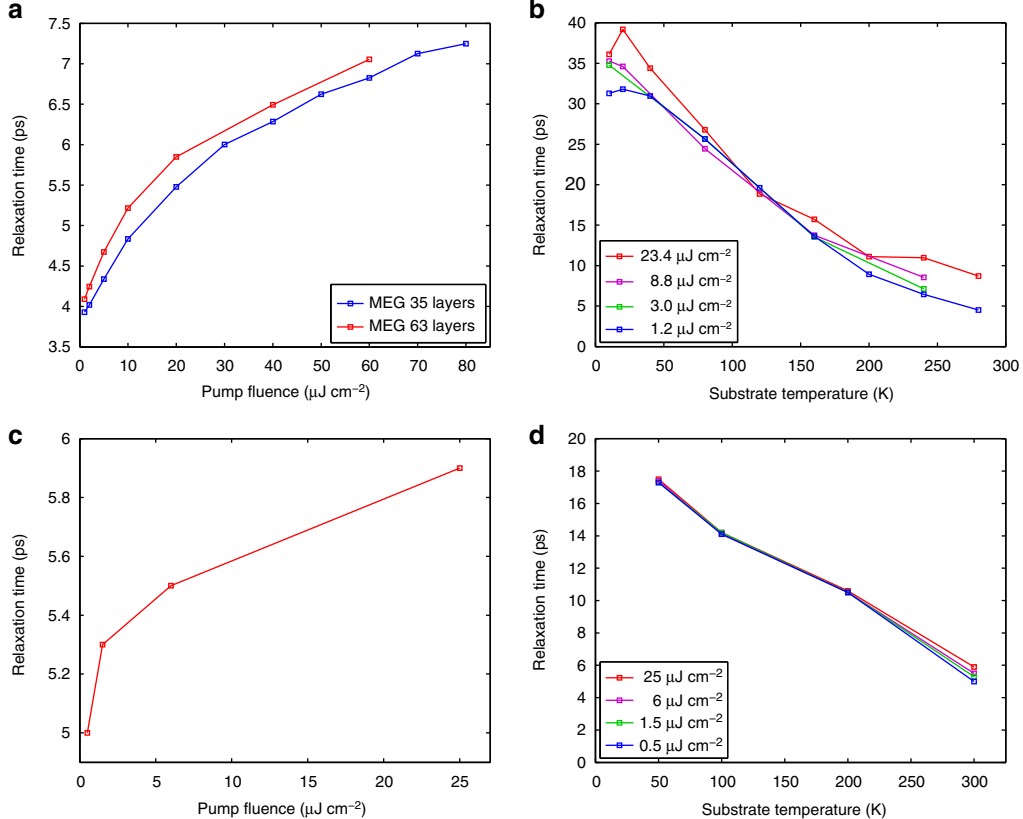

**Figure 5 | THz carrier relaxation times in graphene with low doping density.** (**a**,**b**) Carrier relaxation times extracted from fits to experimental differential THz transmission, $\Delta t/t$, as a function of pump fluence at a substrate temperature of 300 K (**a**) and as a function of substrate temperature for a few different pump fluences (**b**) for lightly doped MEG samples. (**c**,**d**) Carrier relaxation times extracted from fits to theoretical differential THz transmission, $\Delta t/t$, as a function of pump fluence at a substrate temperature of 300 K (**c**) and as a function of substrate temperature for a few different pump fluences (**d**) for disorder-free undoped graphene ($|\varepsilon_F| = 0\,\text{meV}$). The theory accurately reproduces the magnitude of the relaxation times and the trends with pump fluence and substrate temperature observed in the experiments.

Finally, we consider the correlation between the differential THz transmission and the carrier temperature dynamics. In the microscopic theory, the dynamic THz response depends on the transient carrier temperature, the transient Fermi level and the time- and momentum-dependent scattering rates (see Methods section). The inset of Fig. 6b shows a direct comparison between the differential THz transmission and the differential carrier temperature (where the substrate temperature is subtracted) calculated within the microscopic theory for disorder-free undoped graphene ($|\varepsilon_F| = 0\,\text{meV}$) at a substrate temperature of 300 K and a pump fluence of $1.5\,\mu\text{J cm}^{-2}$. We see that the differential THz transmission does not exactly follow the carrier temperature dynamics and, in particular, the relaxation times extracted from the decay of the $\Delta t/t$ signals are not exactly equal to the electronic cooling times. In sharp contrast, a simple Drude model, in which only the carrier temperature is assumed to be time-dependent, would predict incorrectly that the two quantities are proportional[13,14,24]. Hence, our microscopic approach clearly reveals that the transient carrier temperature, the Fermi level shifts and the time- and momentum-dependent scattering rates are all essential to capture the dynamic THz response correctly.

## Discussion

In summary, we have studied the hot-carrier relaxation and cooling dynamics in highly doped and undoped (very lightly doped) graphene samples synthesized using a wide array of methods, using ultrafast time-resolved THz spectroscopy

combined with microscopic modelling. The THz carrier dynamics depend critically on the Fermi level, and are quantitatively explained using a microscopic density-matrix theory of carrier–carrier and carrier–phonon interactions, without the need to invoke any free fitting parameters, phenomenological models or extrinsic effects such as disorder; the theory accounts explicitly for the time-dependent response of the hot carriers to the THz probe field. The hot-carrier dynamics are governed by the interplay of efficient carrier–carrier and carrier–optical–phonon scattering, while carrier–acoustic–phonon scattering is found not to be important on picosecond timescales.

## Methods

**Graphene samples synthesis, fabrication and characterization.** To date a number of techniques have been demonstrated for the synthesis of high-quality graphene including exfoliation, epitaxial and CVD growth methods. We report here comprehensive experiments on graphene synthesized using three different methods. The first type is MEG, which is grown on the C-face of single-crystal 4H-SiC(000$\bar{1}$) substrates by thermal decomposition of Si atoms[34,35]. The MEG grows conformally across atomic terraces on the SiC substrate resulting into large continuous layers with domain sizes exceeding hundreds of micrometres in size. By carefully tuning the chemical recipe, MEG samples having from a few up to a hundred layers can be reliably grown, and the fluctuations in the homogeneity of the MEG samples are estimated not to exceed one layer. The individual layers in MEG are electronically decoupled due to their unique rotational stacking, and each layer exhibits a single graphene layer Dirac cone near the Dirac point, so that MEG behaves in essence as multilayer graphene[48,49]. The second type is sCVDG, which is grown on oxygen-rich copper foil into large individual single crystals exceeding hundreds of micrometres in size[36]. The third type is pCVDG, which is grown on regular copper foil into large continuous layers with domain sizes on the order of hundreds of nanometres[37].

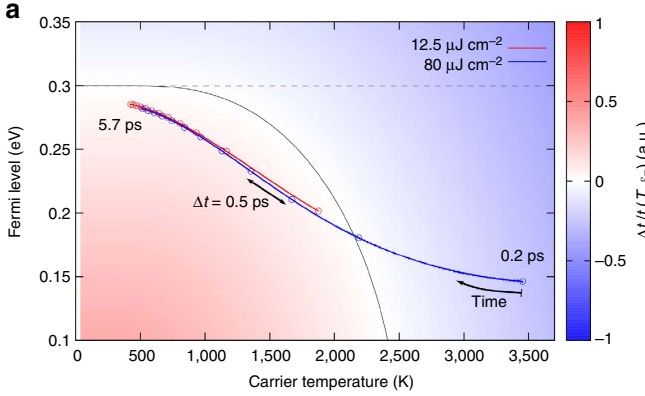

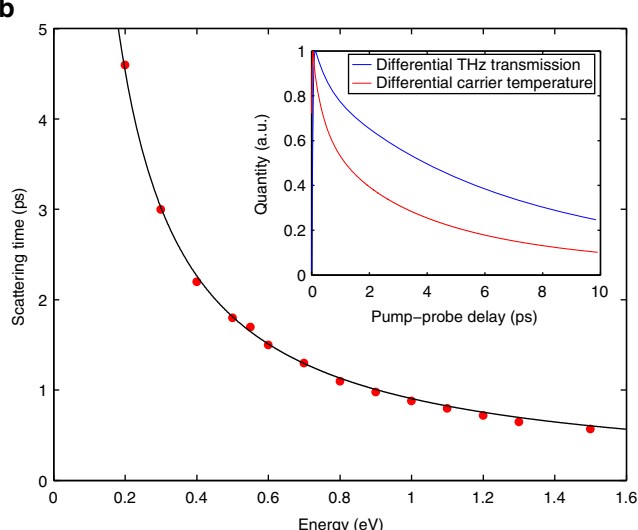

**Figure 6 | Microscopic theory calculation of the carrier dynamics and the carrier scattering time in graphene.** (**a**) Differential THz transmission, $\Delta t/t$, expected from a Drude model with a constant scattering rate for an initial carrier temperature of 300 K and a Fermi level of 300 meV as a function of the transient carrier temperature $T(t)$ and the transient Fermi level $\varepsilon_F(t)$. The solid black line separates regions of positive and negative $\Delta t/t$ signal. The dashed black line shows the possible differential THz transmission under the assumption that the Fermi level remains constant. The solid red (blue) line shows the path through the $T(t)$-$\varepsilon_F(t)$-map for a pump fluence of 12.5 μJ cm$^{-2}$ (80 μJ cm$^{-2}$) obtained from the solution of the full graphene Bloch equations for the pump-induced dynamics. The first point reflects the system 0.2 ps after the optical pump pulse, when the carrier distribution can be represented by a single uniform hot-carrier Fermi-Dirac distribution. The time delay between two points is 0.5 ps, respectively. The figure illustrates the fact that the full microscopic theory is required to explain the experimental data consistently for all photo-excitation levels. (**b**) Carrier scattering time $\tau(\varepsilon)$ as a function of excess carrier energy $\varepsilon$ calculated within the microscopic theory for disorder-free undoped graphene ($|\varepsilon_F| = 0$ meV) at a substrate temperature of 300 K. The calculated values are precisely inverse to the carrier energy, $\tau(\varepsilon) = \beta/|\varepsilon|$ (with $\beta \approx 0.9$ eV ps). Inset: comparison between the normalized differential THz transmission and the normalized differential carrier temperature dynamics calculated within the microscopic theory for disorder-free undoped graphene ($|\varepsilon_F| = 0$ meV) at a substrate temperature of 300 K and a pump fluence of 1.5 μJ cm$^{-2}$. The two time-dependent quantities relax on similar, but not exactly equal timescales.

By carefully tuning the chemical recipe, both mono- and bi-layer pCVDG samples can be reliably grown.

The as-grown CVD graphene samples are transferred from the copper foils to various substrates including C-cut single-crystal sapphire (Alfa Aesar) and amorphous polyethylene (TOPAS cyclic olefin copolymer, TOPAS Advanced

Polymers) for the ultrafast time-resolved THz spectroscopy measurements. The exact same transfer process is used for both sCVDG and pCVDG samples. One side of the copper samples is spin coated with 950PMMA A2 (MicroChem) photoresist as a protection layer and the other side is exposed to oxygen plasma to etch away the undesired graphene. The samples are left in ammonium persulfate solution (0.025 g ml$^{-1}$) for 12 h to dissolve the copper foil underneath. Then, the graphene films with PMMA coating are transferred onto the prepared clean substrates and are left to dry for 12 h. The final step is to use acetone for removing the PMMA coating on top and isopropyl alcohol for rinsing the samples.

Because the different graphene samples are synthesized using completely different techniques, their doping density and degree of disorder are also very different, which allows us to study the dynamic THz response for different Fermi levels and different degrees of disorder. The MEG samples have a gradient doping density profile, where the first few layers closest to the underlying SiC substrate are highly n-doped ($|\varepsilon_F| \sim 100$–400 meV) due to electron transfer from the interface, and the Fermi level in subsequent layers decreases exponentially away from the substrate to around $|\varepsilon_F| \sim 10$ meV (refs 34,35,50,51). The CVDG samples are highly p-doped ($|\varepsilon_F| \sim 200$–400 meV) due to water vapour adsorption from the environment[52]. Thus, we can directly compare the dynamic THz response of graphene with Fermi level far above, far below and very close to the Dirac point.

The graphene degree of disorder is in general not straightforward to quantify, but it can be estimated from various characterization techniques including high-resolution angle-resolved photoemission spectroscopy (ARPES), high-resolution scanning tunnelling microscopy, Raman spectroscopy and electrical transport measurements. Raman spectroscopy measurements of all three types of our graphene samples show negligible D peaks suggesting extremely low disorder[36,37,53,54]. The width of the Dirac cone in the graphene band structure directly measured by ARPES can provide a more sensitive measure for the long-range coherence of graphene[55]. Epitaxial graphene exhibits a very sharp, narrow and well-defined Dirac cone indicating very smooth and homogenous graphene films with extremely high quality. On the other hand, exfoliated and especially CVD graphene transferred to an arbitrary substrate exhibits a rather smeared and broad Dirac cone indicating wrinkled and non-homogenous graphene films. From the width of the Dirac cone, we can extract a correlation length, which for epitaxial graphene exceeds $\sim 50$ nm, limited only by the instrument resolution, but is expected to be much longer[55]. The correlation length for exfoliated and CVD graphene is $\sim 1$–3 nm (ref. 55). A similar disparity in the long-range coherence is inferred also from electrical transport and magneto-optical spectroscopy measurements of the graphene carrier mobility. The carrier mobility of epitaxial graphene has been reported to exceed $\sim 250,000$ cm$^2$ V$^{-1}$ s$^{-1}$ close to the theoretical value for disorder-free graphene[34,35]. On the other hand, the carrier mobilities of exfoliated and CVD graphene transferred to an arbitrary substrate range from a few thousands to tens of thousands dominated by interactions with defects, charge impurities, breaks and ripples. The highest values of up to $\sim 200,000$ cm$^2$ V$^{-1}$ s$^{-1}$ are achieved by minimizing these extrinsic scattering mechanisms including placing the graphene on ultra-smooth substrates such as h-BN[46] or suspending it[47]. Thus, we can also directly investigate the impact of disorder on the dynamic THz response of graphene.

**Ultrafast time-resolved THz spectroscopy on graphene.** To study the dynamic THz response of graphene, we utilize ultrafast time-resolved THz spectroscopy[25,38,39]. Our laser system consists of a Ti:Sapphire oscillator (Mira 900-F, Coherent) followed by a Ti:Sapphire regenerative amplifier (RegA 9050, Coherent) and produces ultrafast optical pulses with a centre wavelength of 800 nm, a pulse width of $\sim 60$ fs and a repetition rate of 250 kHz. A portion of the laser beam is quasi-collimated at the sample position with an intensity spot size diameter of $\sim 1,600$ μm, and optically injects hot carriers in the graphene sample. A second portion of the laser beam illuminates a low-temperature-grown GaAs photoconductive emitter (Tera-SED 3/4, Gigaoptics)[56,57] generating a broadband single-cycle THz pulse which is collimated and focused on the graphene sample with an intensity spot size diameter of $\sim 500$ μm to probe the dynamic THz response. The transmitted portion of the THz probe is detected by using time-domain electro-optic sampling in a 1-mm-thick ZnTe crystal[58–60] and a pair of balanced Si photodiodes. The electrical signal is modulated by a mechanical chopper, placed in either the optical pump or the THz probe arm, and recorded by using a conventional lock-in amplifier data acquisition technique. The graphene sample is mounted inside a liquid helium continuous flow cryostat (ST-100, Janis) to vary the substrate temperature from 10 to 300 K. The time delays between the optical pump, the THz probe and the sampling pulse are controlled by two motorized stages. All THz optics is surrounded by an enclosure purged with purified nitrogen gas to minimize water vapour absorption. The detection bandwidth of the system is in the range of $\sim 0.2$–2.5 THz and the temporal resolution of the measurements is limited by the duration of the THz probe pulse to the sub-picosecond timescale. The experimental error is due primarily to long-term drift of the optomechanical components and the ultrafast Ti:Sapphire laser system, and is estimated not to exceed $\sim 5\%$.

**Microscopic theory calculation of the dynamic THz response of graphene.** The dynamic THz response due to an optical excitation can be microscopically addressed by evaluating the graphene Bloch equations, which describe the coupled

dynamics of the carrier occupation $\rho_{\mathbf{k}}^{\lambda}$ at the wave vector $\mathbf{k}$ in conduction ($\lambda = c$) and valence band ($\lambda = v$), the microscopic polarization $p_{\mathbf{k}}$, and the phonon population $n_{\mathbf{q}}^{j}$ at the momentum $\mathbf{q}$ for different optical and acoustic phonon modes $j$ (ref. 5):

$$\frac{d}{dt}\rho_{\mathbf{k}}^{\lambda} = -\frac{e_0}{\hbar}\mathbf{E}\cdot\nabla_{\mathbf{k}}\rho_{\mathbf{k}}^{\lambda} + 2\Im[\Omega_{\mathbf{k}}^{vc*}p_{\mathbf{k}}] + \Gamma_{\lambda\mathbf{k}}^{\text{in}}[1-\rho_{\mathbf{k}}^{\lambda}] - \Gamma_{\lambda\mathbf{k}}^{\text{out}}\rho_{\mathbf{k}}^{\lambda}, \quad (1)$$

$$\frac{d}{dt}p_{\mathbf{k}} = [i\Delta\omega_{\mathbf{k}} - \gamma_{\mathbf{k}}]p_{\mathbf{k}} - i\Omega_{\mathbf{k}}^{vc}[\rho_{\mathbf{k}}^{c} - \rho_{\mathbf{k}}^{v}]. \quad (2)$$

An optical excitation of the system is considered via the Rabi-frequency $\Omega_{\mathbf{k}}^{vc}$ accounting for interband transitions, where $\Delta\omega_{\mathbf{k}} = v_{\text{F}}k$ is the transition frequency and $v_{\text{F}}$ is the Fermi velocity. Compared to previous work[5], we include a drift term $\mathbf{E}\cdot\nabla_{\mathbf{k}}\rho_{\mathbf{k}}^{\lambda}$ expressing the light-induced intraband transitions, which are crucial for the THz dynamics driven by the probe pulse. The carrier–carrier and carrier–phonon interactions are taken into account by a Boltzmann-like equation with time- and momentum-dependent scattering rates $\Gamma_{\lambda\mathbf{k}}^{\text{in/out}}$ for the carrier occupation and by the diagonal dephasing $\gamma_{\mathbf{k}} = \frac{1}{2}\sum_{\lambda}\Gamma_{\lambda\mathbf{k}}^{\text{in}} + \Gamma_{\lambda\mathbf{k}}^{\text{out}}$ for the microscopic polarization. The explicit form of the many-particle contributions and the equation for the phonon dynamics can be found elsewhere[5,26,27], for example, in ref. 5 in equations 22–24.

The differential THz transmission is given by:

$$\Delta t/t(t,\omega) \propto \alpha^{(t)}(\omega) - \alpha^{(p,t)}(\omega), \quad (3)$$

with the absorption coefficient $\alpha^{(p,t)}(\omega)$ including both the pump and the probe pulse and $\alpha^{(t)}(\omega)$ including only the probe pulse. The absorption $\alpha(\omega) = \Im[j(\omega)/(\epsilon_0\omega^2 A(\omega))]$ is determined by the macroscopic current density[26,27,61]:

$$\mathbf{j}(\omega) = \frac{4e_0\hbar}{m_0 L^2}\sum_{\mathbf{k}}\mathbf{M}_{\mathbf{k}}^{vc}\Im[p_{\mathbf{k}}(\omega)] + \frac{2e_0\hbar}{im_0 L^2}\sum_{\mathbf{k},\lambda}\mathbf{M}_{\mathbf{k}}^{\lambda\lambda}\rho_{\mathbf{k}}^{\lambda}(\omega), \quad (4)$$

where $\mathbf{M}_{\mathbf{k}}^{\lambda\lambda'}$ is the optical matrix element, $m_0$ is the free electron mass and $L^2$ is the structure area of the system that cancels out after performing the summation over $\mathbf{k}$. The current contains an interband contribution that is driven by the microscopic polarization $p_{\mathbf{k}}(\omega)$ and an intraband contribution that is determined by the carrier occupation $\rho_{\mathbf{k}}^{\lambda}(\omega)$. While the interband term has the dominant contribution for probe pulses at optical frequencies, we consider in this work the intraband term, which has the dominant contribution for probe pulses at THz frequencies. Assuming a weak THz probe pulse, $\rho_{\mathbf{k}}^{\lambda}(\omega)$ can be treated perturbatively:

$$\rho_{\mathbf{k}}^{\lambda}(t) = \rho_{\mathbf{k}}^{\lambda,0}(t) + \delta\rho_{\mathbf{k}}^{\lambda}(t), \quad (5)$$

where $\rho_{\mathbf{k}}^{\lambda,0}(t)$ is the pump-induced carrier occupation, while $\delta\rho_{\mathbf{k}}^{\lambda}(t)$ describes the weak carrier occupation excited by the THz probe pulse.

To obtain the dynamic THz response from the differential THz transmission spectra, we exploit the fact that ultrafast carrier–carrier scattering in graphene forms a uniform hot-carrier Fermi-Dirac distribution within the first tens of femtoseconds after the excitation[3,6,7] and the subsequent dynamics is fully characterized by the temporal evolution of the transient carrier temperature $T(t)$ and the transient Fermi level $\varepsilon_{\text{F}}(t)$. Thus, by iteratively evaluating $T(t)$ and $\varepsilon_{\text{F}}(t)$ on the basis of the numerically calculated carrier dynamics at each time step, we obtain the pump-induced carrier occupation $\rho_{\mathbf{k}}^{\lambda,0}(t)$. For the dynamics of the probe-induced carrier occupation, we derive from equation (1) and with the ansatz in equation (5) a separate equation of motion yielding:

$$\frac{d}{dt}\delta\rho_{\mathbf{k}}^{\lambda}(t) = -\frac{e_0}{\hbar}\mathbf{E}\cdot\nabla_{\mathbf{k}}\rho_{\mathbf{k}}^{\lambda,0}(t) - \Gamma_{\lambda\mathbf{k}}^{0}(t)\delta\rho_{\mathbf{k}}^{\lambda}(t), \quad (6)$$

where $\Gamma_{\lambda\mathbf{k}}^{0}(t) = \Gamma_{\lambda\mathbf{k}}^{\text{in},0}(t) + \Gamma_{\lambda\mathbf{k}}^{\text{out},0}(t)$ is the diagonal contribution stemming from the Boltzmann-like scattering terms (equation (1)) which are independent of the probe pulse as denoted by the index 0. Non-linear contributions in the probe pulse and non-diagonal terms have been neglected here. Finally, we numerically evaluate the Fourier transform of $\delta\rho_{\mathbf{k}}^{\lambda}(t)$ from equation (6) and, thereby, we account for the fully microscopically determined carrier dynamics in terms of $\rho_{\mathbf{k}}^{\lambda,0}(t)$ and $\Gamma_{\lambda\mathbf{k}}^{0}(t)$. Thus, our microscopic approach provides access to the Coulomb- and phonon-assisted dynamics induced by the THz probe pulse and is used to obtain the results shown in Figs 2–5.

**Microscopic theory calculation of the carrier scattering time in graphene.** Here, we derive an analytic relation between the microscopic scattering rates and the exponential decay of the carrier occupation. The collision part of the Boltzmann equation, which is given by:

$$\frac{d}{dt}\rho_{\mathbf{k}}^{\lambda} = \Gamma_{\lambda\mathbf{k}}^{\text{in}}[1-\rho_{\mathbf{k}}^{\lambda}] - \Gamma_{\lambda\mathbf{k}}^{\text{out}}\rho_{\mathbf{k}}^{\lambda}, \quad (7)$$

can be rewritten as:

$$\frac{d}{dt}\rho_{\mathbf{k}}^{\lambda} = -\frac{\rho_{\mathbf{k}}^{\lambda} - \tau_{\lambda\mathbf{k}}\Gamma_{\lambda\mathbf{k}}^{\text{in}}}{\tau_{\lambda\mathbf{k}}}, \quad (8)$$

where we define:

$$\tau_{\lambda\mathbf{k}} := \frac{1}{\Gamma_{\lambda\mathbf{k}}^{\text{in}} + \Gamma_{\lambda\mathbf{k}}^{\text{out}}}. \quad (9)$$

The term $\tau_{\lambda\mathbf{k}}\Gamma_{\lambda\mathbf{k}}^{\text{in}}$ can be written as:

$$\tau_{\lambda\mathbf{k}}\Gamma_{\lambda\mathbf{k}}^{\text{in}} = \frac{\Gamma_{\lambda\mathbf{k}}^{\text{in}}}{\Gamma_{\lambda\mathbf{k}}^{\text{in}} + \Gamma_{\lambda\mathbf{k}}^{\text{out}}} = \frac{1}{1 + \frac{\Gamma_{\lambda\mathbf{k}}^{\text{out}}}{\Gamma_{\lambda\mathbf{k}}^{\text{in}}}}. \quad (10)$$

By accounting only for weak excitations, the scattering rates can be approximated with the equilibrium scattering rates $\Gamma_{\lambda\mathbf{k}}^{\text{in/out},0}$ fulfilling the principle of detailed balance (D.B.):

$$\frac{\Gamma_{\lambda\mathbf{k}}^{\text{out}}}{\Gamma_{\lambda\mathbf{k}}^{\text{in}}} \approx \frac{\Gamma_{\lambda\mathbf{k}}^{\text{out},0}}{\Gamma_{\lambda\mathbf{k}}^{\text{in},0}} \overset{\text{D.B.}}{=} e^{(\varepsilon_k - \varepsilon_{\text{F}})/k_{\text{B}}T}. \quad (11)$$

Thus, equation (10) represents the initial Fermi distribution $\rho_{\mathbf{k}}^{\lambda,0} \approx \tau_{\lambda\mathbf{k}}\Gamma_{\lambda\mathbf{k}}^{\text{in}}$ and the Boltzmann equation yields the relaxation-time model:

$$\frac{d}{dt}\rho_{\mathbf{k}}^{\lambda} = -\frac{\rho_{\mathbf{k}}^{\lambda} - \rho_{\mathbf{k}}^{\lambda,0}}{\tau_{\lambda\mathbf{k}}}. \quad (12)$$

Within the approximation in equation (11), equation (12) clearly reveals a direct relation between the microscopic scattering rates and the exponential decay of the carrier occupation. However, by fitting the numerical data stemming from the full scattering equation (equation (7)), our method for the determination of the carrier scattering time (see Fig. 6b) goes beyond the relaxation-time approximation.

**Drude model as an approximation of the microscopic theory.** To obtain a simple Drude model from our microscopic approach (equations (1)–(6)), we approximate the scattering rate as a constant: $\Gamma_{\lambda\mathbf{k}}^{0}(t) = \Gamma$, that is, we neglect the time and the momentum dependence. By assuming a $\delta$-shaped perturbation for the THz probe pulse, the Fourier transform of equation (6) yields:

$$\delta\rho_{\mathbf{k}}^{\lambda}(\omega) = i\omega\mathbf{e}_x\frac{e_0 A_0}{\hbar}\cdot\frac{1}{i\omega - \Gamma}\nabla_{\mathbf{k}}\rho_{\mathbf{k}}^{\lambda,0}. \quad (13)$$

The intraband current density is given by:

$$\mathbf{j}(\omega) = \frac{2e_0\hbar}{im_0 L^2}\sum_{\mathbf{k},\lambda}\mathbf{M}_{\mathbf{k}}^{\lambda\lambda}\delta\rho_{\mathbf{k}}^{\lambda}(\omega). \quad (14)$$

By assuming a uniform hot-carrier Fermi-Dirac distribution for the carrier occupation, the gradient yields:

$$\nabla_{\mathbf{k}}\rho_{\mathbf{k}}^{\lambda} = -\sigma_{\lambda}\mathbf{e}_{\mathbf{k}}\frac{v_{\text{F}}\hbar}{4k_{\text{B}}T}\text{sech}^2[\sigma_{\lambda}(kv_{\text{F}}\hbar - \varepsilon_{\text{F}})/2k_{\text{B}}T], \quad (15)$$

where $\sigma_{\lambda}$ is 1 ($-1$) for $\lambda = c$ ($\lambda = v$) and $\varepsilon_{\text{F}}$ is the Fermi level. For a constant scattering rate $\Gamma$ and with the optical intraband matrix element $\mathbf{M}_{\mathbf{k}}^{\lambda\lambda} \approx i\sigma_{\lambda}M\mathbf{e}_{\mathbf{k}}$ ($\sigma_c = 1$ and $\sigma_v = -1$), the intraband current can be evaluated analytically using equations (13) and (14):

$$j(\omega) = \frac{e_0^2 MA_0 k_{\text{B}}}{m_0\pi c^2}\left[i\frac{\Gamma}{\Gamma^2 + \omega^2} - \frac{\omega}{\Gamma^2 + \omega^2}\right]\times T\left[\ln\left(1 + e^{\varepsilon_{\text{F}}/k_{\text{B}}T}\right) + \ln\left(1 + e^{-\varepsilon_{\text{F}}/k_{\text{B}}T}\right)\right], \quad (16)$$

which corresponds to the Drude model. For a constant scattering rate $\Gamma$ and an arbitrary THz frequency, the resulting Drude-like differential THz transmission (see Fig. 6a) is given by:

$$\Delta t/t(t) \propto j(T_0, \varepsilon_{\text{F},0}) - j(T(t), \varepsilon_{\text{F}}(t)), \quad (17)$$

where $T_0$ and $\varepsilon_{\text{F},0}$ denote the initial carrier temperature and Fermi level before the arrival of the optical pump pulse at time $t = 0$.

We note that in principle a $\mathbf{k}$-dependent $\Gamma_{\mathbf{k}}$ could also be considered in equations (13) and (14) to obtain a more advanced Drude model. However, this would require also an approximate analytical model for $\Gamma_{\mathbf{k}}$. We again emphasize that we go beyond the approximation by using the numerically calculated microscopic scattering rates for the evaluation of the current.

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

## Acknowledgements

We appreciate Xiaohan Wang and Rodney S. Ruoff for providing the single-crystal CVD graphene (sCVDG) samples. M.T.M. and C.J.D. thank Steve Katnik for technical assistance with the ultrafast laser system. F.K., T.W., E.M. and A.K. thank Roland Jago (Research Training Group 1558) for fruitful discussions. The work at the University of Michigan and the Georgia Institute of Technology was supported in part by the National Science Foundation (NSF) Materials Research Science and Engineering Center (MRSEC) under grant DMR-0820382. The work at the University of Michigan was also supported in part by the NSF Center for Photonic and Multiscale Nanomaterials (CPHOM) under grant DMR-1120923 and the NSF CAREER Award (ECCS-1254468). T.W. and A.K. acknowledge partial financial support from the Deutsche Forschungsgemeinschaft (DFG) through SPP-1459 and Sfb 787. E.M. acknowledges partial financial support from the European Commission (EC) under the Graphene Flagship programme (contract No. CNECT-ICT-604391) and the Swedish Research Council (VR). C.B. acknowledges partial financial support from the EC under the Graphene Flagship programme (contract No. CNECT-ICT-604391). This work was performed in part at the Lurie Nanofabrication Facility (LNF), a member of the National Nanotechnology Infrastructure Network (NNIN), which is supported in part by the NSF.

## Author contributions

M.T.M. and C.J.D. performed and analysed the ultrafast time-resolved THz spectroscopy experiments. F.K., T.W., E.M. and A.K. developed the microscopic theory. S.L., C.-H.L. and Z.Z. provided the polycrystalline CVD graphene (pCVDG) samples. C.B. and

W.A.d.H. provided the multilayer epitaxial graphene (MEG) samples. M.T.M. and T.B.N. with input from other authors wrote the paper. All authors discussed the results.

## Additional information

**Competing financial interests:** The authors declare no competing financial interests.

