## [Peer review file · Nature Communications]

Transferred manuscripts:

Reviewers' Comments:

Reviewer #1 (Remarks to the Author)

In my first report, I confronted the Authors with a long list of technical questions about their manuscript. The Authors have thoroughly answered all my questions with extended and informative answers. I see no reason why this manuscript should not be published in Nature Communications.

Reviewer #2 (Remarks to the Author)

I believe that the authors have addressed all comments from previous reviews in a satisfactory way and the paper can be published.

Response to Reviewers:

We thank you for considering and accepting our manuscript entitled "Microscopic Origins of the Terahertz Carrier Relaxation and Cooling Dynamics in Graphene" (NCOMMS-16-03106-T) for publication in Nature Communications. We also thank the referees for their hard work and helpful comments. We have revised our manuscript to address any final critical points and to ensure that it complies with the journal format requirements. We trust that, with these final editorial and formatting changes, the manuscript will be suitable for publication.

Reviewer #1 (Remarks to the Author):

Comment: In my first report, I confronted the Authors with a long list of technical questions about their manuscript. The Authors have thoroughly answered all my questions with extended and informative answers. I see no reason why this manuscript should not be published in Nature Communications.

Response: We thank the reviewer for his/her comment.

Reviewer #2 (Remarks to the Author):

Comment: I believe that the authors have addressed all comments from previous reviews in a satisfactory way and the paper can be published.

Response: We thank the reviewer for his/her comment.